# Evaluating the Effects of Biochar with Farmyard Manure under Optimal Mineral Fertilizing on Tomato Growth, Soil Organic C and Biochemical Quality in a Low Fertility Soil

**Iqra Rehman** [1], **Muhammad Riaz** [1,*], **Sajid Ali** [2], **Muhammad Saleem Arif** [1], **Shafaqat Ali** [1,3,*], **Mohammed Nasser Alyemeni** [4] and **Abdulaziz Abdullah Alsahli** [4]

1   Department of Environmental Sciences & Engineering, Government College University Faisalabad, Faisalabad 38000, Pakistan; meerabfatima7@gmail.com (I.R.); msarif@outlook.com (M.S.A.)
2   Institute of Agricultural Science, Quaid-i-Azam Campus, The University of Punjab, Lahore 54590, Pakistan; sajid.iags@pu.edu.pk
3   Department of Biological Sciences and Technology, China Medical University, Taichung 40402, Taiwan
4   Department of Botany and Microbiology, College of Science, King Saud University, Riyadh 11451, Saudi Arabia; mnyemeni@ksu.edu.sa (M.N.A.); aalshenaifi@ksu.edu.sa (A.A.A.)
*   Correspondence: mr548@ymail.com (M.R.); shafaqataligill@gcuf.edu.pk (S.A.)

**Abstract:** Biochar amendments are widely recognized to improve crop productivity and soil biogeochemical quality, however, their effects on vegetable crops are less studied. This pot study investigated the effects of cotton stick, corncob and rice straw biochars alone and with farmyard manure (FYM) on tomato growth, soil physico–chemical and biological characteristics, soil organic carbon (SOC) content and amount of soil nutrients under recommended mineral fertilizer conditions in a nutrient-depleted alkaline soil. Biochars were applied at 0, 1.5 and 3% ($w/w$, basis) rates and FYM was added at 0 and 30 t ha$^{-1}$ rates. Biochars were developed at 450 °C pyrolysis temperature and varied in total organic C, nitrogen (N), phosphorus (P) and potassium (K) contents. The results showed that biochars, their amounts and FYM significantly improved tomato growth which varied strongly among the biochar types, amounts and FYM. With FYM, the addition of 3% corncob biochar resulted in the highest total chlorophyll contents (9.55 ug g$^{-1}$), shoot (76.1 cm) and root lengths (44.7 cm), and biomass production. Biochars with and without FYM significantly increased soil pH, electrical conductivity (EC) and cation exchange capacity (CEC). The soil basal respiration increased with biochar for all biochars but not consistently after FYM addition. The water-extractable organic C (WEOC) and soil organic C (SOC) contents increased significantly with biochar amount and FYM, with the highest SOC found in the soil that received 3% corncob biochar with FYM. Microbial biomass C (MBC), N (MBN) and P (MBP) were the highest in corncob biochar treated soils followed by cotton stick and rice straw biochars. The addition of 3% biochars along with FYM also showed significant positive effects on soil mineral N, P and K contents. The addition of 3% corncob biochar with and without FYM always resulted in higher soil N, P and K contents at the 3% rate. The results further revealed that the positive effects of biochars on above-ground plant responses were primarily due to the improvements in below-ground soil properties, nutrients' availability and SOC; however, these effects varied strongly between biochar types. Our study concludes that various biochars can enhance tomato production, soil biochemical quality and SOC in nutrient poor soil under greenhouse conditions. However, we emphasize that these findings need further investigations using long-term studies before adopting biochar for sustainable vegetable production systems.

**Keywords:** biochar; soil organic C; nutrient availability; soil biochemical quality; tomato

## 1. Introduction

Biochar is an organic amendment containing stable organic carbon (C) and is produced from biomass waste through pyrolysis in the absence of oxygen under a wide variety

of pyrolysis conditions. Biochar is considered a low cost and eco-friendly amendment showing positive effects on soil and plant productivity [1–5]. In addition to increasing soil organic C and mitigating climate change, biochar has widely been recognized for its ability to improve soil physical, chemical and biological properties [6–11]. For example, biochar can improve soil structure, porosity and water-holding capacity [12], alter pH, electrical conductivity (EC), cation exchange capacity (CEC) and soil organic C contents [13–15], and improve soil biological functions including microbial biomass, extra-cellular enzymatic activities and microbial communities [16]. These effects of biochar on soil properties regulate soil nutrients' retention and availabilities to plants [7]. However, these effects of biochar on soil-plant systems vary strongly with biochar types and characteristics, i.e., feedstocks and pyrolysis conditions (temperature and time duration) [17]. Moreover, biochar is considered to be relatively stable than their feedstock biomass in soil [18]. However, the properties of biochar amendments can also undergo changes after their application to soils [19,20].

Biochar improves soil mineral nitrogen (N) contents by retaining ammonium ($NH_4^+$) and nitrate ($NO_3^-$), and reducing their losses to environment through leaching and gaseous emissions [21,22]. Biochar can also enhance soil phosphorus (P) availability and retention from mineral and organic sources [23]. Recent studies have indicated that biochar has potential to increase soil potassium (K) availability to plants [24]. As a result, biochar can improve N, P and K use efficiencies [25,26]. Earlier studies have demonstrated that biochar with chemical fertilizers enhanced crop growth, root traits and grain yield by improving soil fertility [27–29]. However, application of biochar with other organic amendments showed stronger potential for improving soil fertility, soil organic C and crop yields than its sole application [1,30,31]. Biochar as a component of compost can have synergistic benefits by enhancing microbial activity and reducing nutrient losses during composting [32]. Similary, biochar with other organic amendments improved plant growth and reduced N and P losses to environment [33]. Bonanomi et al. [34] argued that addition of biochar with other organic amendments can result in both synergistic and antagonist effects on soil–plant systems depending on the nature of co-applied organic amendments. However, effects of biochar on agricultural crops and vegetables have yielded different results. Despite of the positive effects of biochar on crop yields, various meta-analysis studies have reported contrasting findings e.g., Crane-Droesch et al. [35] reviewed 84 studies to report that crop yield increased after initial application of biochar but effects were more pronounced in highly weathered soils than in organic matter rich soils, Jeffery et al. [36] concluded that biochar enhanced only tropical crop yields, and Ye et al. [37] found that biochar increased crop yields by 15% but only with mineral fertilizers. However, the majority of these studies investigated cereals crops and effects of biochar on vegetable crops are less reported, e.g., [38,39]. A review by Singh et al. [40] reported 10 studies on the effects of biochar on vegetables and found that nine of these studies showed that biochar enhanced vegetable growth, yield and water use efferences. Biochar enhanced tomato growth and soil fertility but did not affect the quality and yield [41].

Tomato is a common crop with huge economic values and a widespread global delivery. Moreover, tomato production in greenhouses has increased rapidly because it plays a vital role in delivering fresh tomatoes to the consumers [42]. Therefore, such intensive tomato production systems require further research to optimize the factors such as the usage of soil organic amendments, fertilizers and environmental conditions. In Pakistan, during 2011, tomato was grown on an area of 52,300 hectares with an average production of 9.5–10.5 tons per hectare [6]. We hypothesized that biochar would increase the growth of tomatoes by improving soil physico–chemical and biochemical properties, nutrient availability and soil organic C; however, these effects will vary with biochar types, and addition of FYM with biochars would provide further additive benefits. The objectives of the present study, therefore, were to evaluate the effects of cotton stick, corncob and rice straw biochars, their amounts (0, 1.5% and 3%) and FYM (0 and 30 t ha$^{-1}$) on growth of

tomatoes, soil properties, nutrient availability and soil organic C under optimal mineral fertilizer conditions in nutrient poor alkaline soil.

## 2. Material and Methods

### 2.1. Collection of Soil, Feedstocks and FYM

The soil for the greenhouse pot study was collected from an agriculture field under wheat cultivation near Lahore (31.5204° N, 74.3587° E). Soil samples were taken from 0 to 20 cm depth with an augur, air dried and passed through a 2.0-mm sieve. The sieved soil was mixed thoroughly to get homogenized composite soil sample. Manure was collected from the livestock form, airdried and passed through a 2.0-mm sieve. The cotton stick, corncob and rice straw feedstocks were collected from an agricultural farm, and were air-dried for several days before pyrolysis.

### 2.2. Preparation and Characterization of Biochar

Biochar was produced by pyrolyzing the feedback biomass in a locally fabricated kiln [25]. The pyrolysis was conducted at 450 °C for 45 min and the peak pyrolysis temperature was achieved at 7–10 °C min$^{-1}$ heating rate. After preparation, biochar was crushed to pass through a 5.0-mm mesh to obtain a uniform particle size. Total organic C and total N contents of biochar were determined on a Vario Micro CHNS-O Analyzer (Elementar Analysensysteme GmbH, Hanau, Germany). For the determination of total P and K contents, biochar samples were digested in $H_2O_2$ and $H_2SO_4$ solutions following the method of Wolf [43]. Total P contents in supernatants were estimated following a vanadate–molybdate method on a UV visible spectrophotometer (Dynamica Halo DB-20 Series, Livingston, UK) whereas K contents were measured on a flame photometer having 0.2 ppm detection limit (Jenway, Cole-Parmer, St. Neots, UK).

### 2.3. Pot Experiment and Treatments

The pot study was performed under greenhouse conditions at the Institute Agricultural Sciences, University of the Punjab, Lahore (31.4790° N, 74.2662° E). In total, 2.5 kg of soil was weighed into 3 L plastic pots to which treatments were mixed thoroughly following a completely randomized (CRD) experimental design under factorial arrangements and each treatment had four replicates. The three biochars i.e., cotton stick, corncob and rice straw were applied at 0, 1.5 and 3% rates ($w/w$, basis) along with 0 and 30 t ha$^{-1}$ FYM. Considering the 1.51 t m$^{-3}$ bulk density of the soil, the 1.5 and 3.0% rate of biochar was equivalent to 34 and 68 t ha$^{-1}$, respectively. A total of 10 seeds of Roma Local tomato cultivar were sown and, after germination, they were thinned to three healthy plants in each pot. The pots were irrigated with tap water to keep the soil moistened at 60% water holding capacity. The mineral N, P and K were applied at 75, 60 and 60 kg ha$^{-1}$ rate, respectively using urea, diammonium phosphate (DAP) and muriate of potash (MOP) after emergence. The mineral fertilizers were dissolved in 50-mL tap water and applied as liquid fertilizer. The pots were kept in a glass greenhouse for 45 days and any weeds that emerged were manually removed.

### 2.4. Plant Harvest and Analysis

After 45 days, plants were irrigated to facilitate uprooting and harvesting under moist conditions. In total, two out of three harvested plants were used to record morphological attributes and the remaining plant was used to measure chlorophyll contents. The harvested plants were cut into shoot and root parts to measure shoot and root lengths, and record fresh shoot and root weights. After taking the data from the fresh plants, they were oven-dried at 70 °C for over-night in a fan-forced oven (Eyela WFO-600ND, Tokyo Rikakikai, Tokyo, Japan) until constant weight to measure oven-dry shoot and root weights.

The leaves of fresh plants were washed in distilled water and dried by placing them in the layer of filter papers before being processed for chlorophyll content determination. Briefly, 0.2 g fresh leaf was ground in 10 mL of 80% acetone using a pestle and mortal.

The extracts were filtered through Whatman#1 filter paper by giving two to three 5-mL washings with 80% acetone to get a final volume of 25 mL. Concentrations of chlorophyll *a*, *b* and total chlorophyll in the extracts were estimated by measuring absorbance of samples at 663 and 645 nm, respectively, on a UV Spectrophotometer (Dynamica Halo DB-20 Series, UK) [44].

*2.5. Soil Analysis*

2.5.1. Initial Analysis

Soil pH and electrical conductivity (EC) were measured on 1:5 ratio (*w/v*, basis) in water with a pre-calibrated pH (InoLab, WTW Series, Germany) and EC meter (Jenway, UK) [45]. Soil texture and particle size analysis were performed following the hydrometer method [46]. Soil bulk density (BD) was determined using the core method [47]. Cation exchange capacity (CEC) was estimated from the concentrations of exchangeable cations ($Na^+$, $K^+$, $Ca^{2+}$ and $Mg^{2+}$) which were measured as the soluble salts in solution from washing 5 g soil with 25 mL of 60% ethanol [48]. The extractable and exchangeable $Na^+$, $K^+$, $Ca^{2+}$ and $Mg^{2+}$ concentrations in the soil extracts were analyzed on an Inductively Coupled Plasma Atomic Emission Spectroscopy (ICP-AES, Agilent, Santa Clara, CA, USA). Soil organic C (SOC) contents were determined with the wet acid-sulphate digestion method of Walkley and Black [49,50]. Total N was measured using the Kjeldahl's digestion and distillation method [51], whereas soil-available N was estimated following the method described by Øien and Selmer-Olsen [52]. Soil total P content was analyzed using the vanadate–molybdate method of Chapman and Pratt [53] and soil-available P content was determined following the method described by Kue [54]. Soil-available K contents were estimated following a flame photometer method.

2.5.2. Post-Experimental Soil Analysis

After the plant harvest, moist soil samples from each pot were taken, freed from any live and/or dead vegetation parts, homogenized before transferring them into resealable plastic bags and kept at 4 °C before analysis. Soil moisture contents were determined gravimetrically by over-drying the samples at 105 °C for overnight until constant weight in a fan-forced oven (Eyela WFO-600ND, Tokyo Rikakikai, Tokyo, Japan). Soil pH and EC were measured on moist samples whereas CEC and SOC were determined on oven-dried samples followed the methods described earlier.

Soil basal respiration was measured with the alkali trap method (Alef, 1995). Briefly, 3 g oven-dry equivalent moist soil sample was placed into 500 mL mason jars containing 10 mL 1 M NaOH solution vials to capture the $CO_2$ evolved. After incubation for 1 week in the dark at 25 °C, the $CO_2$ absorbed by NaOH solution was estimated by titrating against 0.1 M HCl after the addition of $BaCl_2$. Water extractable organic C (WEOC) content in moist soil samples was measured following the procedure of Ghani et al. [55]. A total of 5 g of moist soil was transferred into 50-mL plastic vials, of which 25 mL of distilled water was added before the suspension was shaken on a horizontal shaker at 150 rpm for 90 min and then centrifuged at $5000 \times g$ for 10 min. The resultant supernatant was filtered through a Whatman#42 filter paper. The total organic C content in extracts was determined using the Walkley and Black method.

Microbial biomass C (MBC), microbial biomass N (MBN) and microbial biomass P (MBP) were measured following the chloroform fumigation extraction method [56–58]. In total, 20 g of oven-dried equivalents of field moist soil samples were weighed in duplicate and the first set of the samples was fumigated with ethanol-free chloroform for 24 h at 25 °C in a desiccator. The second non-fumigated set was divided into two subsets of 10-g to be extracted with 40 mL 0.5 M $K_2SO_4$ and 0.5 M $NaHCO_3$, respectively, by shaking the mixtures at 150 rpm for 60 min and filtering the suspensions through Whattmann#42 filter papers. After fumigation, the first set of samples was also divided into two 10-g portions which were extracted the way similar to non-fumigated samples and the extracts were kept at 4 °C before analysis.

Total organic C contents in non-fumigated and fumigated sample extracts were determined using the modified Walkley Black method [56,59]. Microbial biomass C was then calculated as follows [60]:

$$MBC = K_{EC}/2.64 \tag{1}$$

where $K_{EC}$ is the difference in total organic C contents between the fumigated and non-fumigated samples and 2.64 is the proportionality factor for biomass C released by the fumigation extraction method.

Total N contents in the non-fumigated and fumigated extracts were analyzed following the Kjeldahl digestion method and MBN was calculated as follows [56,61]:

$$MBN = F_N/0.54 \tag{2}$$

where $F_N$ is the difference between total N contents of fumigated and non-fumigated samples and 0.54 is the fractions of biomass N released by fumigation extraction procedure.

The concentrations of inorganic P in $NaHCO_3$ extracts were determined following the ammonium molybdate-ascorbic acid method [62]. The MBN was calculated as difference between the inorganic P contents of fumigated and non-fumigated samples. Moreover, the inorganic P contents of non-fumigated samples were considered as the soil P contents.

For soil ammonium-N ($NH_4$-N) and nitrate-N ($NO_3$-N) analysis, 10-g moist soil samples were extracted with 40-mL 2.0 M KCl solution. The $NH_4$-N concentration in the extracts was determined following the sodium dichloroisocyanurate color reagent method by measuring the absorbance at 660 nm [63]. The $NO_3$-N content was analyzed using the procedure developed by Miranda et al. [64]. The method is based on principle of the nitrate reduction with vanadium (III) followed by the color development from the Griess reaction. Absorbance of the samples was measured at 540 nm. The sum of $NH_4$-N and $NO_3$-N was considered as mineral N contents.

### 2.6. Statistical Analysis

The data were tested for normality to meet the assumptions of analysis of variance (ANOVA) test, and any parameter significantly deviating from normal distribution was log-transformed prior to statistical analysis. A three-way ANOVA was applied to test the effects of biochar types, biochar rates and FYM on tomato growth and soil properties. Tukey's HSD post-hoc test was used to compare multiple treatment means. Figures and tables contain means values of four replicates unless otherwise specified. All statistical analyses were conducted using SPSS for windows software v. 19.

## 3. Results

### 3.1. Properties of Soil, Farmyard Manure and Biochar

The soil used in the experiment was alkaline (pH 8.03, EC 0.16 dS m$^{-1}$) silty clay loam and had low total organic C contents of 12.5 g kg$^{-1}$ (Table 1). The soil is considered critically low in fertility for available N (23.7 mg kg$^{-1}$) and P (3.20 mg kg$^{-1}$) but contain sufficient K contents (85.8 mg kg$^{-1}$). Cotton stick, corncob and rice straw biochars varied in their total organic C, N, P and K contents (Table 2). Total organic C, N, P and K contents in corncob biochar were higher than cotton stick and rice straw biochars. Total organic C, N, P and K contents in FYM were 232, 12.7, 13.4 and 24.6 g kg$^{-1}$, respectively.

**Table 1.** Physical and chemical properties of soil used in the experiment.

| Property | Value |
|---|---|
| pH | 8.03 |
| EC (dS m$^{-1}$) | 0.16 |
| Sand (g kg$^{-1}$) | 467 |
| Silt (g kg$^{-1}$) | 501 |
| Clay (g kg$^{-1}$) | 32 |
| Textural class | Silty clay loam |
| BD (t m$^{-3}$) | 1.50 |
| CEC (cmol$_c$ kg$^{-1}$) | 14.1 |
| Total organic C (g kg$^{-1}$) | 12.5 |
| Total N (g kg$^{-1}$) | 0.61 |
| Total P (g kg$^{-1}$) | 0.24 |
| Total K (g kg$^{-1}$) | 14.3 |
| Available N (mg kg$^{-1}$) | 23.7 |
| Available P (mg kg$^{-1}$) | 3.20 |
| Available K (mg kg$^{-1}$) | 85.8 |

**Table 2.** Total organic C, N, P and K contents in biochars and farmyard manure (FYM).

| Property | Biochar | | | FYM |
|---|---|---|---|---|
| | **Cotton Stick** | **Corncob** | **Rice Straw** | |
| Total organic C (g kg$^{-1}$) | 463 | 732 | 518 | 232 |
| Total N (g kg$^{-1}$) | 11.2 | 13.4 | 10.3 | 12.7 |
| Total P (g kg$^{-1}$) | 4.03 | 6.41 | 2.60 | 13.4 |
| Total K (g kg$^{-1}$) | 16.2 | 17.4 | 21.3 | 24.6 |

*3.2. Effects on Tomato Growth and Physiological Attributes*

The results revealed significant effects of biochar type, biochar rate and FYM on shoot and root lengths of tomato (Figure 1). Both the shoot and root lengths increased with biochar rate and FYM further enhanced shoot and root lengths. The addition of 1.5% and 3% corncob biochar with and without FYM resulted in significantly higher shoot lengths compared to cotton stick and rice straw biochars (Figure 1a). Similar to effects on shoot length, the 1.5% and 3% corncob biochar along FYM also significantly enhanced root lengths (Figure 1b). Fresh shoot and root weights were always the highest with 1.5% and 3% addition of biochars with and without FYM (Table 3). At the 3% corncob biochar addition with FYM, the highest fresh shoot weight of 40.2 g was observed whereas the highest fresh root weight of 4.64 g was noted for the 3% corncob biochar without FYM treatment. With and without FYM, the oven-dry shoot weight of tomato ranged from 0.89 to 3.18, 0.97 to 4.64 and 0.85 to 4.00 g for cotton stick, corncob and rice straw biochars, respectively (Table 3). Similarly, the oven-dry weight with addition of 3% biochar with FYM was also the highest and followed the order corncob (16.2 g) > rice straw (0.76 g) > cotton stick (0.64 g).

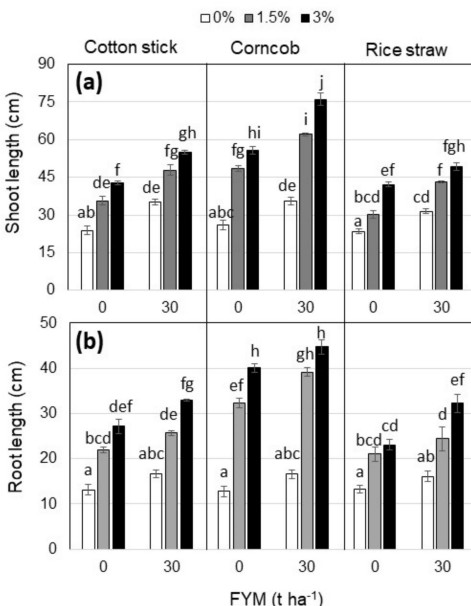

**Figure 1.** Effects of biochar amendments and FYM on (**a**) shoot length (cm) and (**b**) root length (cm) of tomatoes grown for 45 days. Bars represents values of four replicates and contain standard error of means (*n* = 4). Bars with different letters differ significantly from each other at *p* < 0.05.

**Table 3.** Effects of biochar amendments and FYM on fresh and oven-dried shoot and root weights of tomato plants grown for 45 days.

| Biochar Type | FYM (t ha$^{-1}$) | Biochar Rate (%) | Fresh Shoot Weight (g) | Fresh Root Weight (g) | Oven-Dry Shoot Weight (g) | Oven-Dry Root Weight (g) |
|---|---|---|---|---|---|---|
| Cotton stick | 0 | 0 | 9.37 ± 0.75 a | 0.77 ± 0.13 a | 0.89 ± 0.10 a | 0.21 ± 0.04 a |
| | | 1.5 | 13.99 ± 1.25 ab | 1.72 ± 0.09 cd | 1.38 ± 0.10 ab | 0.49 ± 0.02 b–h |
| | | 3 | 20.92 ± 1.93 de | 2.53 ± 0.13 ef | 2.28 ± 0.09 cde | 0.59 ± 0.03 e–h |
| | 30 | 0 | 13.70 ± 1.08 ab | 1.71 ± 0.13 cd | 1.47 ± 0.13 abc | 0.37 ± 0.03 a–d |
| | | 1.5 | 21.87 ± 0.93 de | 2.56 ± 0.16 ef | 2.42 ± 0.14 def | 0.60 ± 0.02 e–h |
| | | 3 | 27.54 ± 0.69 f | 3.57 ± 0.13 gh | 3.18 ± 0.21 fgh | 0.76 ± 0.03 hi |
| Corncob | 0 | 0 | 9.12 ± 0.58 a | 1.70 ± 0.18 cd | 0.97 ± 0.07 a | 0.28 ± 0.05 ab |
| | | 1.5 | 29.92 ± 2.02 fg | 3.82 ± 0.23 h | 2.93 ± 0.14 efg | 0.68 ± 0.03 gh |
| | | 3 | 36.17 ± 1.10 hi | 4.64 ± 0.13 i | 3.37 ± 0.17 gh | 1.00 ± 0.07 j |
| | 30 | 0 | 14.20 ± 1.09 abc | 1.00 ± 0.09 ab | 1.70 ± 0.18 a–d | 0.41 ± 0.03 a–e |
| | | 1.5 | 33.91 ± 1.38 gh | 2.69 ± 0.13 ef | 3.82 ± 0.23 hi | 0.97 ± 0.04 ij |
| | | 3 | 40.24 ± 1.14 i | 3.75 ± 0.19 gh | 4.64 ± 0.13 i | 1.62 ± 0.09 k |
| Rice straw | 0 | 0 | 9.32 ± 0.50 a | 0.85 ± 0.10 ab | 0.85 ± 0.10 ab | 0.23 ± 0.02 a |
| | | 1.5 | 17.94 ± 0.92 bcd | 2.03 ± 0.08 cde | 2.21 ± 0.23 b–e | 0.45 ± 0.02 b–f |
| | | 3 | 21.42 ± 0.40 de | 2.16 ± 0.05 de | 2.72 ± 0.14 efg | 0.50 ± 0.03 |
| | 30 | 0 | 12.70 ± 0.82 ab | 1.45 ± 0.13 bc | 1.66 ± 0.11 a–d | 0.34 ± 0.05 abc |
| | | 1.5 | 19.78 ± 0.82 cd | 3.14 ± 0.06 fg | 2.68 ± 0.26 efg | 0.57 ± 0.04 d–h |
| | | 3 | 26.38 ± 0.66 ef | 2.86 ± 0.04 f | 4.00 ± 0.30 hi | 0.64 ± 0.02 fgh |

Values are means of four replicates followed by ± standard error of means (*n* = 4). In each column, values with different letters differ significantly from each other at *p* < 0.05.

The chlorophyll *a*, *b* and total chlorophyll contents of tomato varied significantly between biochar types, biochar rates and FYM (Table 4). The chlorophyll *a* contents increased with biochar rate and FYM for the cotton stick and corncob biochars; however, this trend was inconsistent under rice straw biochar which reduced chlorophyll *a* with and without FYM application. The chlorophyll *b* contents varied between 2.24–3.60, 2.24–3.92, 2.19–3.27 $\mu$g g$^{-1}$ for cotton stick, corncob and rice straw biochars, respectively. Except for rice straw biochar, the cotton stick and corncob showed pronounced effects on total chlorophyll contents which increased with biochar rate and were higher under FYM. The 3% addition of corncob biochar with FYM resulted in the highest total chlorophyll contents of 9.55 $\mu$g g$^{-1}$.

**Table 4.** Effects of biochar amendments and FYM on chlorophyll *a*, chlorophyll *b* and total chlorophyll contents of tomatoes grown for 45 days.

| Biochar Type | FYM (t ha$^{-1}$) | Biochar Rate (%) | Chlorophyll *a* (ug g$^{-1}$) | Chlorophyll *b* (ug g$^{-1}$) | Total Chlorophyll (ug g$^{-1}$) |
|---|---|---|---|---|---|
| Cotton stick | 0 | 0 | 3.29 ± 0.06 a | 2.24 ± 0.04 a | 5.53 ± 0.07 a |
| | | 1.5 | 3.54 ± 0.12 abc | 2.46 ± 0.03 ab | 6.00 ± 0.10 abc |
| | | 3 | 3.51 ± 0.12 ab | 2.64 ± 0.12 bc | 6.15 ± 0.14 bc |
| | 30 | 0 | 4.46 ± 0.03 efg | 3.16 ± 0.06 fg | 7.62 ± 0.08 fg |
| | | 1.5 | 4.44 ± 0.07 efg | 3.60 ± 0.06 h | 8.04 ± 0.06 gh |
| | | 3 | 5.17 ± 0.08 h | 3.24 ± 0.08 fg | 8.40 ± 0.16 hi |
| Corncob | 0 | 0 | 3.18 ± 0.07 a | 2.24 ± 0.06 a | 5.42 ± 0.07 a |
| | | 1.5 | 3.81 ± 0.07 bcd | 3.10 ± 0.02 de | 6.92 ± 0.08 de |
| | | 3 | 4.01 ± 0.07 cde | 3.45 ± 0.05 gh | 7.46 ± 0.08 efg |
| | 30 | 0 | 4.36 ± 0.15 efg | 3.06 ± 0.08 de | 7.42 ± 0.22 ef |
| | | 1.5 | 4.83 ± 0.03 gh | 3.92 ± 0.04 i | 8.76 ± 0.08 i |
| | | 3 | 5.91 ± 0.05 i | 3.65 ± 0.10 hi | 9.55 ± 0.13 j |
| Rice straw | 0 | 0 | 3.46 ± 0.06 ab | 2.19 ± 0.09 a | 5.65 ± 0.05 ab |
| | | 1.5 | 3.33 ± 0.07 a | 2.41 ± 0.04 ab | 5.74 ± 0.11 abc |
| | | 3 | 3.10 ± 0.08 a | 2.58 ± 0.04 bc | 5.68 ± 0.04 ab |
| | 30 | 0 | 4.52 ± 0.19 fg | 3.19 ± 0.05 fg | 7.71 ± 0.19 gh |
| | | 1.5 | 4.23 ± 0.03 def | 3.27 ± 0.07 fg | 7.50 ± 0.05 efg |
| | | 3 | 3.52 ± 0.14 ab | 2.82 ± 0.06 cd | 6.34 ± 0.19 cd |

Values are means of four replicates followed by ± standard error of means (*n* = 4). In each column, values with different letters differ significantly from each other at *p* < 0.05.

### 3.3. Effects on Soil Physico–Chemical Properties and SOC Dynamics

Changes in soil pH, EC and CEC are shown in Table 5. Soil pH under the corncob biochar was generally less compared to the cotton stick and rice straw biochars. The application of biochars with FYM reduced soil pH than the sole biochars at both rates. Soil EC under cotton stick biochar varied with biochar rate but not after addition of FYM. However, under corncob and rice straw biochars, rate of biochar and FYM increased EC. Moreover, the EC values of 1.46 and 1.49 dS m$^{-1}$ after 3% addition of cotton stick and rice straw with FYM were significantly higher than the similar treatment for the corncob biochar. The soil CEC increased with biochar and FYM for all biochars; however, CEC remained higher under the cotton stick biochar (11.1–42.7 cmolc kg$^{-1}$) compared to corncob (11.4–28.6 cmolc kg$^{-1}$) and rice straw (11.6–33.1 cmolc kg$^{-1}$) biochars (Table 5). Moreover, CEC value after addition of 3% cotton stick with FYM was the highest and significantly different from corncob and rice straw biochars.

**Table 5.** Effects of biochar amendments and FYM on soil pH, electrical conductivity (EC), cation exchange capacity (CEC), basal respiration, water-extractable organic C (WEOC) and soil organic carbon (SOC) contents under tomato grown for 45 days.

| Biochar Type | FYM (t ha$^{-1}$) | Biochar Rate (%) | pH | EC (dS m$^{-1}$) | CEC (cmol$_c$ kg$^{-1}$) | Basal Respiration (mg CO$_2$-C g$^{-1}$ d$^{-1}$) | WEOC (mg kg$^{-1}$) | SOC (g kg$^{-1}$) |
|---|---|---|---|---|---|---|---|---|
| Cotton stick | 0 | 0 | 7.54 ± 0.03 abc | 0.89 ± 0.01 a | 11.1 ± 0.35 a | 0.03 ± 0.00 a | 45.1 ± 2.03 ab | 8.43 ± 0.14 a |
| | | 1.5 | 7.82 ± 0.09 bc | 1.23 ± 0.03 bc | 22.7 ± 0.49 e | 0.12 ± 0.00 c | 64.2 ± 2.37 bcd | 14.9 ± 0.76 c |
| | | 3 | 8.25 ± 0.24 d | 1.45 ± 0.03 d | 31.8 ± 1.32 gh | 0.16 ± 0.00 de | 86.4 ± 2.04 ef | 20.1 ± 0.43 e |
| | 30 | 0 | 7.46 ± 0.02 ab | 1.15 ± 0.01 bc | 15.8 ± 0.64 bc | 0.04 ± 0.00 a | 53.9 ± 1.35 abc | 12.1 ± 0.27 b |
| | | 1.5 | 7.70 ± 0.07 abc | 1.20 ± 0.07 bc | 27.8 ± 0.70 fg | 0.15 ± 0.01 cde | 80.0 ± 2.49 def | 19.6 ± 0.26 de |
| | | 3 | 7.52 ± 0.16 abc | 1.46 ± 0.02 d | 42.7 ± 1.24 i | 0.16 ± 0.01 de | 120 ± 9.25 g | 23.3 ± 0.60 fg |
| Corncob | 0 | 0 | 7.52 ± 0.07 abc | 0.90 ± 0.03 a | 11.4 ± 0.30 a | 0.03 ± 0.00 a | 41.3 ± 2.88 a | 8.93 ± 0.18 a |
| | | 1.5 | 7.63 ± 0.04 abc | 1.09 ± 0.03 b | 15.8 ± 0.55 bc | 0.17 ± 0.01 e | 89.0 ± 2.72 f | 21.6 ± 0.61 ef |
| | | 3 | 7.39 ± 0.03 a | 1.19 ± 0.02 bc | 20.9 ± 0.64 de | 0.23 ± 0.01 f | 129 ± 2.88 gh | 29.9 ± 0.09 h |
| | 30 | 0 | 7.44 ± 0.03 ab | 1.15 ± 0.02 bc | 14.8 ± 0.43 abc | 0.04 ± 0.00 a | 53.2 ± 2.55 abc | 11.8 ± 0.42 b |
| | | 1.5 | 7.39 ± 0.03 a | 1.23 ± 0.01 bc | 24.3 ± 1.46 ef | 0.25 ± 0.01 f | 114 ± 2.72 g | 24.9 ± 0.91 g |
| | | 3 | 7.28 ± 0.03 a | 1.23 ± 0.02 bc | 28.6 ± 1.16 g | 0.34 ± 0.01 g | 145 ± 4.07 h | 35.3 ± 0.22 i |
| Rice straw | 0 | 0 | 7.50 ± 0.03 a | 0.89 ± 0.02 a | 11.6 ± 0.49 ab | 0.03 ± 0.00 a | 46.6 ± 2.27 abc | 8.63 ± 0.15 a |
| | | 1.5 | 7.82 ± 0.02 bcd | 1.18 ± 0.03 bc | 17.1 ± 0.47 cd | 0.09 ± 0.00 b | 68.0 ± 3.39 cde | 12.8 ± 0.25 bc |
| | | 3 | 7.90 ± 0.08 cd | 1.25 ± 0.02 c | 24.2 ± 0.88 ef | 0.12 ± 0.00 c | 85.8 ± 4.34 ef | 17.6 ± 0.32 d |
| | 30 | 0 | 7.43 ± 0.02 ab | 1.13 ± 0.02 bc | 15.2 ± 0.36 abc | 0.04 ± 0.00 a | 55.7 ± 5.27 abc | 11.5 ± 0.48 b |
| | | 1.5 | 7.69 ± 0.06 abc | 1.27 ± 0.02 c | 21.9 ± 0.70 e | 0.12 ± 0.00 c | 82.7 ± 2.54 def | 14.9 ± 0.69 c |
| | | 3 | 7.70 ± 0.06 abc | 1.49 ± 0.04 d | 33.1 ± 0.85 h | 0.13 ± 0.01 cd | 116 ± 3.64 g | 20.9 ± 0.60 ef |

Values are means of four replicates followed by ± standard error of means ($n$ = 4). In each column, values with different letters differ significantly from each other at $p < 0.05$.

The soil basal respiration under corncob biochar was higher and increased with biochar and FYM (Table 5). After application of cotton stick and rice straw biochars, soil basal respiration increased with biochar without any notable influence of FYM. Soil basal respiration without FYM ranged 0.03–0.16, 0.03–0.23 and 0.03–0.12 mg CO$_2$-C g$^{-1}$ d$^{-1}$ for cotton stick, corncob and rice straw biochar, respectively whereas with FYM, soil basal respiration varied from 0.04–0.16, 0.04–034 and 0.04–0.13 mg CO$_2$-C g$^{-1}$ d$^{-1}$, respectively. Similarly, WEOC contents increased significantly with biochar and FYM for all biochars but were higher after addition of 1.5% and 3% corncob biochar compared to cotton stick and rice straw biochars. The addition of 3% corncob biochar with FYM resulted in 81% and 25% higher WEOC than cotton stick and rice straw biochars, respectively (Table 5). The SOC changed significantly in response to biochar and FYM for the three biochars and was 8.43–23.3, 8.93–35.3 and 8.63–20.9 g kg$^{-1}$ under cotton stick, corncob and rice straw biochar, respectively. The differences between the SOC contents under 1.5% and 3% corncob biochar were much larger than cotton stick and rice straw biochars.

### 3.4. Effects on Microbial Biomass

Effects of biochars, their rates and FYM on MBC, MBN and MBP are presented in Figure 2. Without and with FYM, the significantly highest MBC contents were observed under the corncob biochar at both rates followed by cotton stick and rice straw biochars (Figure 2a). The increase in MBC with the addition of 3% corncob biochar along FYM was 17% and 63% higher than cotton stick and rice straw biochars, respectively. The difference in MBC between the 1.5% and 3% biochar addition was also the least for the rice straw biochar. The effects of treatments on MBN were similar to those found for MBC (Figure 2b). The MBN contents were 19.9–64.5, 21.7–81.6 and 21.0–48.7 mg kg$^{-1}$ for cotton stick, corncob and rice straw, respectively. The MBN increased much sharper between the control and application of 1.5% and 3% corncob biochar than cotton stick and rice straw biochars. Addition of 3% corncob biochar along FYM resulted in the significantly highest MBP contents (Figure 2c). Moreover, MBN contents increased with biochar and FYM for all

biochars. The overall MBN contents followed the order: corncob biochar > cotton stick biochar > rice straw biochar.

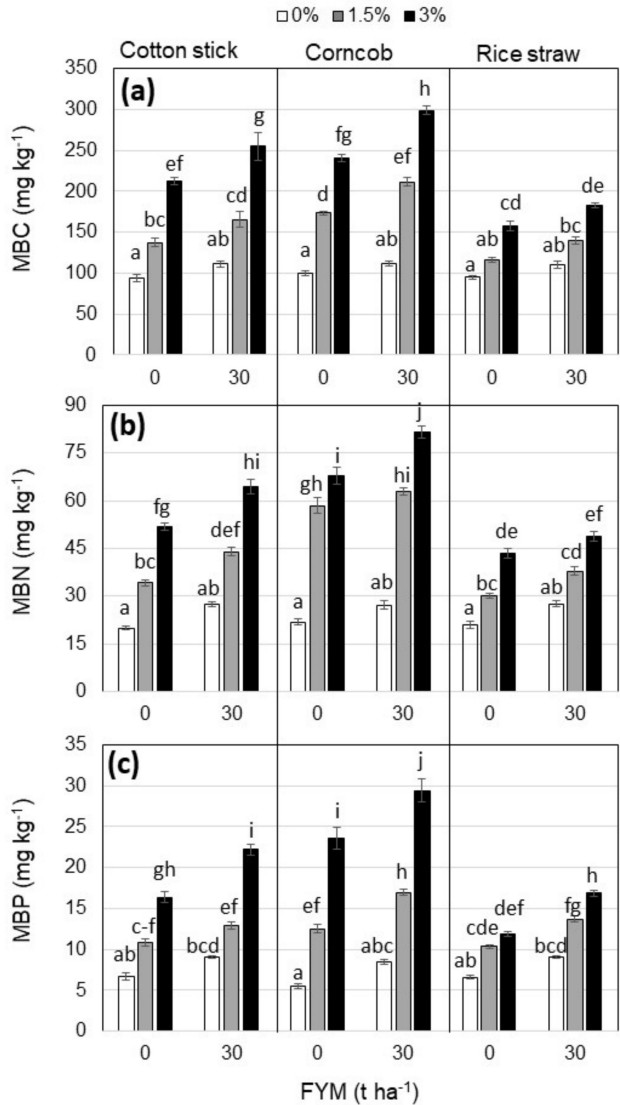

**Figure 2.** Effects of biochar amendments and FYM on soil (**a**) MBC (mg kg$^{-1}$), (**b**) MBN (mg kg$^{-1}$) and (**c**) MBP (mg kg$^{-1}$) under tomatoes grown for 45 days. Bars represents values of four replicates and contain standard error of means (*n* = 4). Bars with different letters differ significantly from each other at *p* < 0.05.

*3.5. Changes in Soil Mineral N, P and K Contents*

The effects of biochars, their rates and FYM on soils $NH_4$-N, $NO_3$-N and mineral N contents are shown in Figure 3. Soil $NH_4$-N contents were 4.20–22.3, 4.69–29.4 and 4.57–17.2 mg kg$^{-1}$ for cotton stick, corncob and rice straw biochars, respectively (Figure 3a). Addition of 1.5% and 3% corncob and cotton stick biochars along FYM showed significantly higher $NH_4$-N contents than rice straw biochar. Soil $NO_3$-N contents, however, were differently affected by biochars, their rates and FYM (Figure 3b). With FYM, the 3% cotton stick biochar increased $NO_3$-N contents in contrast to 1.5% addition which decreased $NO_3$-N contents. For the corncob and rice straw biochars, $NO_3$-N contents increased with biochar and remained higher under the corncob biochar. Soil mineral N contents only increased with the addition of 3% cotton stick biochar without and with FYM (Figure 3c). However, mineral N contents for corncob and rice straw biochars increased with biochar

and FYM, and the highest mineral N contents were found under addition of 3% corncob biochar with FYM.

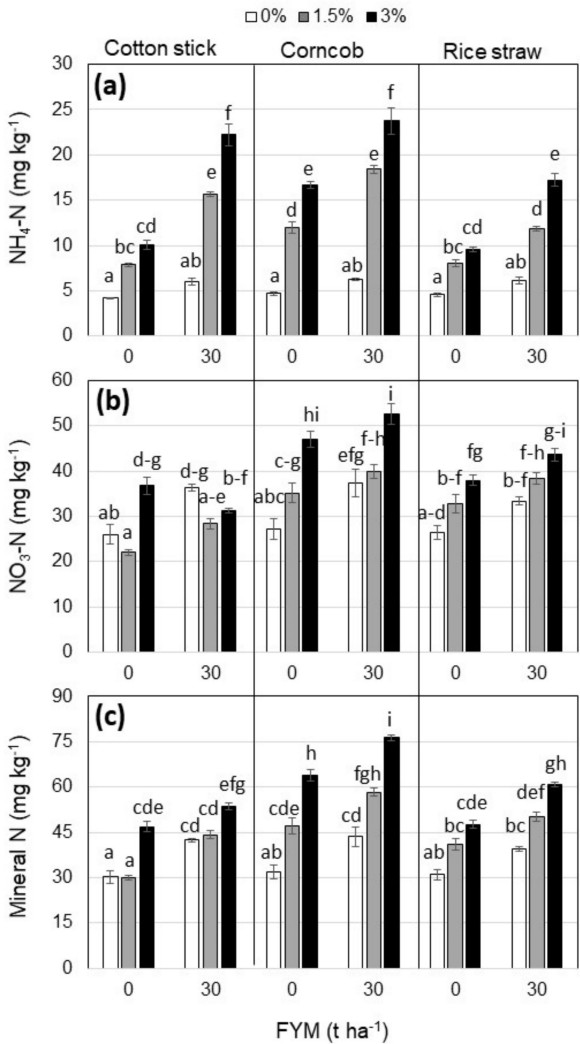

**Figure 3.** Effects of biochar amendments and FYM on soil (**a**) $NH_4$-N (mg kg$^{-1}$), (**b**) $NO_3$-N (mg kg$^{-1}$) and (**c**) mineral N (mg kg$^{-1}$) under tomatoes grown for 45 days. Bars represents values of four replicates and contain standard error of means (*n* = 4). Bars with different letters differ significantly from each other at *p* < 0.05.

Soil P contents were significantly influenced by biochars, their rates and FYM (Table 6). Soil P contents were 4.66–21.2, 4.72–14.4 and 4.78–14.6 mg kg$^{-1}$ for cotton stick, corncob and rice straw biochars, respectively, and FYM always resulted in higher soil P contents for all biochars. Soil K contents also varied significantly with biochars, their rate and FYM (Table 6). Soil K contents increased with rate of biochar for all biochars and FYM further supplemented soil K contents. The highest soil K content of 532 mg kg$^{-1}$ was found under the 3% corncob biochar with FYM treatment which was 23% and 21% higher than the similar treatments under cotton stick and rice straw biochars, respectively.

**Table 6.** Effects of biochar amendments and FYM on soil P and K contents under tomatoes grown for 45 days.

| Biochar Type | FYM (t ha$^{-1}$) | Biochar Rate (%) | Soil P (mg kg$^{-1}$) | Soil K (mg kg$^{-1}$) |
|---|---|---|---|---|
| Cotton stick | 0 | 0 | 4.66 ± 0.46 a | 84.6 ± 3.99 a |
| | | 1.5 | 10.1 ± 0.43 cde | 240 ± 10.1 b |
| | | 3.0 | 16.1 ± 0.50 h | 420 ± 14.2 efg |
| | 30 | 0 | 7.03 ± 0.30 ab | 113 ± 11.2 a |
| | | 1.5 | 12.4 ± 0.42 efg | 273 ± 6.59 bc |
| | | 3.0 | 21.2 ± 0.49 i | 431 ± 19.9 fg |
| Corncob | 0 | 0 | 4.72 ± 0.31 a | 76.4 ± 4.40 a |
| | | 1.5 | 8.04 ± 0.21 bc | 323 ± 8.89 cd |
| | | 3.0 | 12.0 ± 0.73 ef | 358 ± 7.78 def |
| | 30 | 0 | 6.52 ± 0.30 ab | 119 ± 5.84 a |
| | | 1.5 | 10.7 ± 0.48 de | 423 ± 17.2 efg |
| | | 3.0 | 14.4 ± 0.38 fgh | 532 ± 43.21 h |
| Rice straw | 0 | 0 | 4.78 ± 0.24 a | 85.1 ± 3.49 a |
| | | 1.5 | 8.84 ± 0.45 bcd | 253 ± 13.1 bc |
| | | 3.0 | 11.4 ± 0.74 e | 371 ± 11.1 ef |
| | 30 | 0 | 7.01 ± 0.40 ab | 131 ± 8.28 a |
| | | 1.5 | 11.5 ± 0.30 e | 350 ± 4.84 de |
| | | 3.0 | 14.6 ± 0.78 gh | 440 ± 7.61 g |

Values are means of four replicates followed by ± standard error of means ($n$ = 4). In each column, values with different letters differ significantly from each other at $p < 0.05$.

## 4. Discussion

Our study compares the effects of cotton stick, corncob and rice straw biochars, applied at 0, 1.5% and 3% rate without and with FYM, on growth of tomatoes, soil biogeochemical properties and soil organic C contents. We found that all biochars promoted tomato growth and FYM further augmented the biochars' effects. Without and with FYM, effects of biochars were strongly rate-dependent and varied between biochars types, and the overall data showed that the corncob biochar was more beneficial followed by cotton stick and rice straw biochars. The positive effects of biochar on tomato growth and quality, i.e., plant height, root length and stem diameter has been reported earlier which increased with biochar amount [65,66]. Moreover, biochar along optimum irrigation at higher rate enhanced tomato growth and development [67]. Agbna et al. [68] also found the positive effects of biochar on tomatoes from germination to maturity due to modification in growth environment. Biochars significantly increased chlorophyll *a*, chlorophyll *b* and total chlorophyll contents of tomatoes in our study. Chlorophyll molecules are essential for harvesting light energy in photosynthesis, and for all physiological responses. Chlorophyll *a* and *b* molecules have distinct characteristics and varying chemical structures which enable them to absorb specific infrared light by acting as photoreceptors to perform photosynthetic functions. However, relatively limited research has shown the positive effects of various biochars on growth and yield of some vegetable crops e.g., stems of Lantana camarai biochar on okra [69]; pine wood biochar on cucumber [70]; birch wood biochar on potato [71]; and, maize straw biochar on tomato [72] and pumpkin [73]. These findings imply that the effects of biochar varied strongly with their types and rates, soil conditions and crops.

Changes in soil physico–chemical characteristics from biochar application closely influence crop growth and yield patterns. We found that the effects of biochars on soil

pH and EC were significant and varied with biochar type and rate, and FYM application. pH decreased after addition of corncob biochar and FYM but increased with cotton stick and rice straw biochars, and remained less under FYM. Changes in soil pH can alter soil chemical environment to influence nutrient availability to plants [74]. Due to the general alkaline nature of various biochars, they increase soil pH of acidic soils [75]. Many studies have showed that pH of biochars were 5.9–12.3 (mean value 8.9) whereas others had pH range of 6.2–9.6 (mean value 8.1) [76,77]. In this study, biochars with FYM resulted in minor and not-always significant changes in soil EC. Soil EC reflects the concentrations of solvable salts which affect soil nutrients and their availability to plants. Biochar can alter soil EC; however, many studies have also reported the use of biochar to ameliorate negative effects of soil salinity on plants [78–80]. Application of biochar added with ash having solvable salts can result in minor improvements in EC [80]. We observed minor changes in soil pH and EC under biochars and FYM; however, these changes exerted non-deleterious effects on tomato growth and soil nutrients. Arif et al. [25] also reported such minor changes in soil pH and EC after addition of *Acacia* biomass-derived biochar to maize crop which did not impose any negative effects on crop growth and nutrient availability. The effects on soil CEC, however, were more consistent and varied significantly between biochars, their rates and FYM in this study. The CEC increased with biochars and FYM and the highest CEC values of 42.7 and 33.1 $cmol_c$ $kg^{-1}$ were found after addition of 3% cotton stick and rice straw biochars with FYM, respectively. Different biochars can induce variable effects on soil CEC based on their characteristics such as the feedstock nature and pyrolysis conditions [81]. For example, biochar developed at a low temperature can have higher CEC and a potential role in nutrient availability and soil fertility [82,83]. Barring a few studies, the majority of studies have reported positive effects of biochar on soil CEC [82]. The presence of carboxyl groups and oxidation of aromatic hydrocarbons in biochar could increase soil CEC in biochar-amended soil and influence soil nutrient availability [7,84].

The effects of biochar amendments on soil microbial activity and C dynamics, i.e., C mineralization, WEOC and SOC are generally pronounced because of the inevitable interactions of biochar with native and fresh soil organic matter [85]. The soil C mineralization, measured as soil basal respiration, increased with biochars, and was the highest under the corncob biochar along FYM. This could be due to higher labile C contents in corncob biochar resulting in higher microbial activity and C mineralization [85,86]. However, the lower soil basal respiration for the cotton stick and rice straw biochars without notable influence of FYM might suggest the presence of less labile than recalcitrant C and varying degree of microbial co-metabolism for C substrate from soil, biochar and FYM [87]. The WEOC contents generally increased with biochars and FYM but was significantly higher for the corncob biochar. The biochar-specific changes in WEOC contents could be the result of differences in C adsorption which controls C retention and use by microbes [88,89]. Therefore, the effects on microbial biomass (MBC, MBN and MBP) were similar to those on WEOC in this study. The MBC values reflect the ability of microorganisms to access C from biochar and FYM because C substrate and other nutrient can be adsorbed to biochar surface [87]. The higher MBC in the combined biochar and FYM treatments suggested higher microbial accessibility of WEOC which could have resulted from desorption and degradation of biological matter and C substrate from biochar. However, the differences in MBC contents between biochars were probably due to variations in microbial ability to access soluble C that influence MBC and net C mineralization [85,86]. The contrasting effects of biochars on soil basal respiration and MBC might be due to the potential of biochar to adsorb soil native and added C (FYM) which increased the MBC but had little effects on soil basal respiration [90]. A meta-analysis by Zhou et al. [91] argued that the increase in soil MBC under biochar amendments depended on soil conditions rather than biochar characteristics. Biochar can decrease soil C mineralization by adsorbing nutrients, WEOC and microorganisms on its surface leading to higher C use efficiency and low activities of C mineralizing enzymes [92–94]. This could also explain the higher SOC contents under biochars and FYM which increased with biochar amount and remained the highest under the corncob biochar.

Biochar-led increases in MBN and MBP in our study were mediated by characteristics of biochars than the soil [91]. The stimulation in MBC, MBN and MBP from biochar can have positive implications for SOC, N and P retention in soils [95].

Our study showed that biochars significantly enhanced soil mineral N ($NH_4$-N and $NO_3$-N), P and K contents, and the effects varied with biochars and their addition rates. The reduction in $NO_3$-N contents under cotton stick biochar with FYM could be due to N integration into MBN confirmed from higher MBN contents than the corresponding treatments without FYM [96]. Our findings are consistent with Agegnehu et al. [97] who reported higher soil $NH_4$-N and $NO_3$-N contents from willow wood biochar and found that manure further enhanced soil mineral N contents. The higher mineral N retention by biochar can simultaneously increase N availability to plants by reducing the risk of N pollution to the environment from leaching and $N_2O$ emissions [22,98]. The mineral N contents of soil in excess of plant requirements can undergo nitrification and denitrification to produce $N_2O$ [99] and biochar can reduce $N_2O$ emissions by altering these processes [100]. However, these effects can vary substantially with biochar nature and soil conditions e.g., He et al. [101] reported decreased $N_2O$ emissions from rice straw biochar by reinstating nitrification in Oxisols whereas, in contrast, He et al. [102] reported higher $N_2O$ and $NH_3$ emissions under wheat straw biochar. The positive effects of biochar on soil P contents in our study are similar to those observed by Borchard et al. [103] who showed that biochar increased soil P contents by decreasing leaching losses in sandy and loamy soils. The higher soil P contents in the combined biochar and FYM can also indicate more P availability to plants [97]. Our results about the effects of biochar on soil K availability are consisting with previous studies e.g., Amin [104] reported significant positive effects of corncob biochar on soil K availability and wheat growth in alkaline calcareous soil whereas Qayyum et al. [105] found that various straw-based biochars enhanced K availability, cotton growth and yield in nutrient poor soil. In addition to affecting K availability from soil, mineral fertilizer and FYM, the higher K contents of biochar might also have increased soil K contents in this study [93,105]. Higher nutrient availabilities from biochar amendments are primary factor leading to better crop growth and yields.

We found that application of biochars with FYM was more beneficial in improving tomato growth, soil biogeochemical characteristics, nutrients availability and soil organic C. The benefits of organic amendments on soil fertility, plant nutrient uptake and crop growth are widely recognized [106–108]. Our study is consistent with recent research that biochar with raw and composted-manure result in higher biomass production and crop yields [25,109]. Agegnehu et al. [110] reviewed the role of biochar and biochar-compost mixtures on soil quality and crop yield, and reported nearly 20% increase in crop yields at about 10 t ha$^{-1}$ biochar rate. However, our findings contradict those of Bass et al. [111] who showed that combining biochar with compost improved soil properties but not always increased crop yields compared to sole applications of biochar and compost.

## 5. Conclusions

This pot study on the effects of three biochars (cotton stick, corncob, rice straw), biochar rates (0, 1.5% and 3%) and FYM (0, 30 t ha$^{-1}$) on tomato growth, soil physico–chemical and biological properties, soil nutrient and soil organic C demonstrated positive effects of biochars on tomato productivity and soil quality under greenhouse conditions. All biochars improved tomato productivity, soil biochemical properties, soil nutrients and SOC; however, the corncob biochar appeared to be more effective than cotton stick and rice straw biochars. The addition of FYM further complemented the biochars' ability to improve tomato production and soil quality. The observed higher soil organic C, N, P and K contents along the concomitant increase in MBC, MBN and MBP suggest the potential of biochar amendments to conserve C, N and P contents in nutrient poor soil while improving plant productivity. Our study, however, emphasizes conducting more long-term studies to further optimize the use of biochars in vegetable production systems considering the costs and benefits to the environment and society.

**Author Contributions:** Conceptualization, I.R. and M.R.; methodology, I.R., M.R. and S.A.; formal analysis, I.R. and M.R.; investigation, I.R.; resource, M.R. and S.A. (Sajid Ali); writing—original draft preparation, I.R. and M.R.; writing—review and editing, M.R., S.A. (Sajid Ali), M.S.A., S.A. (Shafaqat Ali), M.N.A. and A.A.A.; visualization, I.R and M.R.; supervision, M.R. and S.A.; project administration, M.R.; funding acquisition, M.R., S.A. (Shafaqat Ali) and M.N.A. All authors have read and agreed to the published version of the manuscript.

**Funding:** The study was also supported by Higher Education Commission (HEC) Pakistan NRPU#20-3485 (M. Riaz), HEC Indigenous PhD Fellowship (I. Rehman) and King Saud University, Riyadh, Saudi Arabia Researchers Supporting Project Number RSP-2020/180 (M. N. Alyemeni).

**Informed Consent Statement:** Not applicable.

**Data Availability Statement:** Not applicable.

**Acknowledgments:** The authors highly acknowledge the Government College University, Faisalabad, Pakistan for its support. The authors would like to extend their sincere appreciation to the Researchers Supporting Project Number (RSP-2020/180), King Saud University, Riyadh, Saudi Arabia.

**Conflicts of Interest:** The authors declare that they have no conflict of interest.

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
