# Peer review of "Evaluating the Effects of Biochar with Farmyard Manure under Optimal Mineral Fertilizing on Tomato Growth, Soil Organic C and Biochemical Quality in a Low Fertility Soil"

_sustainability, doi:10.3390/su13052652_

Round 1

Reviewer 1 Report

General comments:

Manuscript deals with "soil amendment with using biochar".

Biochar is low cost, environmentally friendly, and can be applied for a variety of purposes [https://doi.org/10.3390/w11030551], and it could enhance the fertility of soil, and organism growth by supplying nutrients and immobilising organisms on its surface [https://doi.org/10.3390/microorganisms9010004].

1. Page 1, Line 19: "..soil nutrients and soil organic carbon (SOC).." Should be corrected as "... soil organic carbon (SOC) and nutrients...".

2. Page 1, Line 25; "... higher chlorophyll contents, shoot and root lengths..." Mention to their values.

3. Page 1, Line 26; "Biochars and FYM significantly altered soil pH...." Increased or decreased? mention to their values.

Author Response

Manuscript deals with "soil amendment with using biochar".

Comment: Biochar is low cost, environmentally friendly, and can be applied for a variety of purposes [https://doi.org/10.3390/w11030551], and it could enhance the fertility of soil, and organism growth by supplying nutrients and immobilising organisms on its surface [https://doi.org/10.3390/microorganisms9010004].

Response: Thank you for suggesting to add this information in the manuscript. These studies are now included in the revised parts of the introduction section.

Comment 1. Page 1, Line 19: "..soil nutrients and soil organic carbon (SOC).." Should be corrected as "... soil organic carbon (SOC) and nutrients...".

Response: Corrected

Comment 2. Page 1, Line 25; "... higher chlorophyll contents, shoot and root lengths..." Mention to their values.

Response: The sentence is revised to add the values for these parameters.

Comment: 3. Page 1, Line 26; "Biochars and FYM significantly altered soil pH...." Increased or decreased? mention to their values.

Response: The sentence is corrected

Reviewer 2 Report

This manuscript described the effects of different types of biochars on the tomato growth, and the corresponding biochemical effects, as well as the effect of co-composing farmyard manure. Many different parameters were evaluated in this work. However, this work is barely well written with lots of flaws. Some sentences are really hard to read and understand, some are verbose and should be simplified, an overall revision on the language by a native speaker is highly suggested. Therefore, I would suggest the authors making revision on this manuscript with the following comments to be addressed: 

1) Delete the paragraph 1 in the introduction. There is no connection between Paragraph 1 and the next several paragraphs.  

2) Line 83. The authors mentioned that there are only 10 studies shows enhancement, then it would be better if the authors provide the numbers of studies that shows the opposite effects. 

3) In the whole manuscript, there is no superscript, nor subscripts. The authors should find a way to revise the whole manuscript. 

4) Unit confusion. Suggest revise the whole manuscript with the International Standard Unit System. 

5) Line 125. “1.51Mg m-3”? what does it mean? 1.51 ton? 

6) Why chose to measure chlorophyll contents? What does it stand for? 

7) The method of measuring microbial biomass. What’s the results if this method was applied for biochars, FYM and soil? 

8) The title of 3.2 and 3.2 is the same, why? 

9) Table 5, the Basel respiration value. Does it represent the rate per day, or per 45day? 

10) Line 450, should be “not always increase crop yield” 

11) For all the tables and figures, the authors should mention the duration that these values were obtained. 

Author Response

Comment: This manuscript described the effects of different types of biochars on the tomato growth, and the corresponding biochemical effects, as well as the effect of co-composing farmyard manure. Many different parameters were evaluated in this work. However, this work is barely well written with lots of flaws. Some sentences are really hard to read and understand, some are verbose and should be simplified, an overall revision on the language by a native speaker is highly suggested. Therefore, I would suggest the authors making revision on this manuscript with the following comments to be addressed:

Response: Thank you for this very critical comment and suggestion which has helped us to improve the overall quality of the manuscript. We have revised manuscript at various places to remove typographic and grammatical mistakes. Moreover, our colleague proficient in English has also gone through the manuscript.

Comment 1): Delete the paragraph 1 in the introduction. There is no connection between Paragraph 1 and the next several paragraphs.

Response: We agree with the reviewer and have deleted the sentence.  

Comment 2): Line 83. The authors mentioned that there are only 10 studies shows enhancement, then it would be better if the authors provide the numbers of studies that shows the opposite effects.

Response: The review article by Singh et al. (2019) found only 10 studies relating to the effects of biochar on vegetable production and 9 of these studied found positive effects. We have corrected the statement now.  

Comment 3): In the whole manuscript, there is no superscript, nor subscripts. The authors should find a way to revise the whole manuscript.

Response: We apologies for this mistake occurred while formatting the manuscript. We have corrected this throughout the manuscript.

Comment 4): Unit confusion. Suggest revise the whole manuscript with the International Standard Unit System.

Response: Revised throughout the manuscript.

Comment 5): Line 125. “1.51Mg m-3”? what does it mean? 1.51 ton?

Response: This value referred to soil bulk density and its SI units are Mg m-3.  

Comment 6): Why chose to measure chlorophyll contents? What does it stand for?

Response: Chlorophyll molecules are important components of plant leaves which absorb light energy and converts it into chemical energy through photosynthesis, and thus reflect changes in physiological responses of plants towards treatments. We have provided this information in the discussion section of the manuscript.

Comment 7): The method of measuring microbial biomass. What’s the results if this method was applied for biochars, FYM and soil?

Response: We used the chloroform fumigation extraction method for the determination of microbial biomass C, N and P which is widely used method. The method is unequivocally useful for soils under variety of amendments such as biochars and FYM.

Comment 8): The title of 3.2 and 3.2 is the same, why?

Response: It was a typographic mistake and is corrected now. We have corrected figure and tables captions throughout the manuscript.

Comment 9): Table 5, the Basel respiration value. Does it represent the rate per day, or per 45 day?

Response: Yes, soil basel respiration was measured by incubating moist soils for 24 hours. Therefore, the unit is mg CO2-C g-1 d-1 which is now corrected in Table 5.

Comment 10): Line 450, should be “not always increase crop yield”

Response: Corrected.

Comment 11): For all the tables and figures, the authors should mention the duration that these values were obtained.

Response: The experiment was lasted for 45 days and after this period, measurements on plant and soil samples were made. We have inserted this information in captions to all figures and tables.

Round 2

Reviewer 2 Report

Line 125. “1.51Mg m-3” should be 1.51 ton m-3

Author Response

Comment: Line 125. “1.51 Mg m-3” should be 1.51 ton m-3

Response: We have corrected the units